# Diversity of Coordination Modes in a Flexible Ditopic Ligand Containing 2-Pyridyl, Carbonyl and Hydrazone Functionalities: Mononuclear and Dinuclear Cobalt(III) Complexes, and Tetranuclear Copper(II) and Nickel(II) Clusters †

**Evangelos Pilichos [1], Evangelos Spanakis [1], Evangelia-Konstantina Maniaki [1], Catherine P. Raptopoulou [2] , Vassilis Psycharis [2],* , Mark M. Turnbull [3],* and Spyros P. Perlepes [1,4],***

[1] Department of Chemistry, University of Patras, 26504 Patras, Greece
[2] Institute of Nanoscience and Nanotechnology, NCSR "Demokritos", 15310 Aghia Paraskevi Attikis, Greece
[3] Carlson School of Chemistry and Biochemistry, Clark University, Worcester, MA 01610, USA
[4] Institute of Chemical Engineering Sciences, Foundation for Research and Technology-Hellas (FORTH/ICE-HT), Platani, B.O. Box 1414, 26504 Patras, Greece
* Correspondence: v.psycharis@inn.demokritos.gr (V.P.); MTurnbull@clarku.edu (M.M.T.); perlepes@patreas.upatras.gr (S.P.P.); Tel.: +30-210-6503346 (V.P.); +1-508-7937167 (M.M.T.); +30-2610-996730 (S.P.P.)
† This article is dedicated to Professor Masahiro Yamashita—a great scientist and a precious friend—on the occasion of his 65th birthday.

**Abstract:** Syntheses, crystal structures and characterization are reported for four new complexes $[Cu_4Br_2(L)_4]Br_2$ (**1**), $[Ni_4(NO_3)_2(L)_4(H_2O)](NO_3)_2$ (**2**), $[Co_2(L)_3](ClO_4)_3$ (**3**) and $[Co(L)_2](ClO_4)$ (**4**), where $L^-$ is the monoanion of the ditopic ligand *N′*-(**1**-(pyridin-2-yl)ethylidene)pyridine-2-carbohydrazide (LH) built on a picolinoyl hydrazone core fragment, and possessing a bidentate and a tridentate coordination pocket. The tetranuclear cation of **1**·0.8H₂O·MeOH is a strictly planar, rectangular [2 × 2] grid. Two 2.21011 $L^-$ ligands bridge adjacent $Cu^{II}$ atoms on the short sides of the rectangle through their alkoxide oxygen atoms, and two 2.11111 ligands bridge adjacent $Cu^{II}$ atoms on the long sides of the rectangle through their diazine groups; two metal ions are 5-coordinate and two are 6-coordinate. The tetranuclear cation of **2**·0.2H₂O·3EtOH is a square [2 × 2] grid. The two 6-coordinate $Ni^{II}$ atoms of each side of the square are bridged by the alkoxide O atom of one 2.21011 $L^-$ ligand. The dinuclear cation of **3**·0.8H₂O·1.3MeOH contains two low-spin octahedral $Co^{III}$ ions bridged by three 2.01111 $L^-$ ligands forming a *pseudo* triple helicate. In the mononuclear cation $[Co(L)_2]^+$ of complex **4**, the low-spin octahedral $Co^{III}$ center is coordinated by two tridentate chelating, meridional 1.10011 ligands. The crystal structures of the complexes are stabilized by a variety of π–π stacking and/or H-bonding interactions. Compounds **2**, **3** and **4** are the first structurally characterized nickel and cobalt complexes of any form (neutral or anionic) of LH. The 2.01111 and 1.10011 coordination modes of $L^-$, observed in the structures of complexes **3** and **4**, have been crystallographically established for the first time in coordination complexes containing this anionic ligand. Variable-temperature, solid-state dc magnetic susceptibility and variable-field magnetization studies at 1.8 K were carried out on complexes **1** and **2**. Antiferromagnetic metal ion···metal ion exchange interactions are present in both complexes. The study reveals that the cation of **1** can be considered as a practically isolated pair of strongly antiferromagnetically coupled (through the diazine group of $L^-$) dinulear units. The susceptibility data for **2** were fit to a single-*J* model for an *S* = 1 cyclic tetramer. The values of the *J* parameters have been rationalized in terms of known magnetostructural correlations. Spectral data (infrared (IR), ultraviolet/visible (UV/VIS), ¹H nuclear

magnetic resonance (NMR) for the diamagnetic complexes) are also discussed in the light of the structural features of **1**–**4** and the coordination modes of the organic and inorganic ligands that are present in the complexes. The combined work demonstrates the ligating flexibility of L$^-$, and its usefulness in the synthesis of complexes with interesting structures and properties.

**Keywords:** coordination clusters; [2 × 2] grids; magnetic studies; *N'*-(1-(pyridin-2-yl)ethylidene) pyridine-2-carbohydrazide cobalt(III); nickel(II) and copper(II) complexes

---

## 1. Introduction

The word "ligand" is derived from the Latin verb "*ligare*" meaning "to bind" [1]. It was first introduced by Alfred Stock when lecturing in Berlin (1916) on the chemistry of boranes and silanes. However, it came into common use only in the 1950s, mainly through the PhD Thesis of Jannik Bjerrum [2]. Nowadays, the appropriate use of old ligands and the design of new, sophisticated ones is one of the pylons of modern inorganic chemistry. Theoretical concepts related to ligands include the chelate effect, the macrocyclic effect, the conformation of chelating rings, the chemistry of non-innocent ligands, the hard and soft bases concept and the isoelectronic and isolobal relationships, among others. Of particular interest is also the study of the reactivity of coordinated ligands, an approach in which the metal ion activates a proligand, transforming it through an in situ reaction and providing unusual ligands that sometimes cannot be synthesized by conventional organic or inorganic synthesis [3,4]. The proper choice of bridging ligands has played a key role in the development of modern magnetochemistry and the interdisciplinary field of molecular magnetism [5], where the metal···metal exchange interactions mediated through the bridges are responsible for a variety of interesting magnetic phenomena [6–9].

Polytopic organic ligands are particularly interesting in coordination chemistry and magnetochemistry. Their design and subsequent synthesis introduces preprogrammed coordination information that is "stored" in the coordination pockets [10,11]. When such ligands react with a transition metal ion, it interprets this information according to its own coordination "algorithm". If the coordination pocket does not contain many donor atoms to fully saturate the coordination requirements of the metal ion, self-assembly can take place favoring the formation of homoleptic or heteroleptic coordination clusters [12]. The obtained nuclearity depends largely on the polytopic nature of the ligand and the preferred metal ion coordination number and geometry [10–13]. This in turn leads to a wide variety of magnetic exchange interactions, which depend on the number and nature of bridges and the magnetic orbitals that are available.

The ligand of the present work is *N'*-(1-(pyridin-2-yl)ethylidene)pyridine-2-carbohydrazide [other names: methyl(pyridin-2-yl)methanone picolinoylhydrazone or 2-acetylpyridine picolinoylhydrazone], drawn in its enol-imino form in Scheme 1 and abbreviated as LH. It is a ditopic ligand built on a picolinoyl hydrazone core fragment (it can also be considered as an asymmetric alkoxy diazine ligand [14]) possessing a bidentate and a tridentate coordination pocket. The deprotonated ligand (L$^-$) has two potentially bridging functional groups (μ-O, μ-N-N) and, because of the free rotation around the N–N single bond, can exist in two different coordination conformers, both of which can in principle form spin-coupled dinuclear and polynuclear metal complexes with quite different magnetic properties. We decided to work with this ligand because its published coordination chemistry has been limited [14–21]. Since no Co(II) and Ni(II) complexes of L$^-$ have been reported, we first targeted compounds with these metal ions. We were also interested in preparing Cu(II) complexes, because the only reported complex [Cu$_4$(L)$_4$(H$_2$O)$_2$](NO$_3$)$_4$ [14] is a structurally impressive square [2 × 2] grid and can be considered magnetically as an essentially isolated pair of antiferromagnetically coupled dinuclear fragments. We report herein our results from the synthetic investigation of the CuBr$_2$/LH, Ni(NO$_3$)$_2$·6H$_2$O/LH and Co(ClO$_4$)$_2$·6H$_2$O/LH reaction systems and the characterization of the products

obtained. This paper can be considered as a continuation of our interest in the chemistry and magnetism of 3d-metal coordination clusters [9,22], and in the coordination and metal ion-meditated/promoted transformation properties of polydentate ligands containing two or more functionalities (including 2-pyridyl, carbonyl and hydrazone/azine groups, among others) [4,23–27].

LH

**Scheme 1.** The free ligand *N′*-(1-(pyridin-2-yl)ethylidene)pyridine-2-carbohydrazide (LH) drawn in its enol-imino form.

## 2. Results and Discussion

### 2.1. Synthetic Comments

A variety of $M^{II}/X^-/LH/B$ (M = Co, Ni, Cu; X = Cl, Br, $NO_3$, $ClO_4$; B = $Et_3N$, LiOH, $R_4NOH$, $NaO_2CR'$ with R,R′ = various groups) reaction systems, involving various solvent media, reagent ratios and crystallization techniques, were systematically investigated before arriving at the optimized synthetic conditions reported in Section 3. In many instances we have isolated microcrystalline powders with reasonable analytical data, but we report here only the structurally characterized products.

The $CuBr_2/NaO_2CPh/LH$ (1:1:1) reaction mixture in MeOH gave a green solution from which greenish brown crystals of $[Cu_4Br_2(L)_4]Br_2 \cdot 0.8H_2O \cdot MeOH$ ($\mathbf{1} \cdot 0.8H_2O \cdot MeOH$) were subsequently isolated in a good yield (~60%). Assuming that **1** is the only product from the reaction system, its formation can be summarized by Equation (1). Use of other bases, e.g., $Et_3N$ and $Me_4NOH \cdot 5H_2O$, gave powders of the same product (infrared (IR) evidence).

$$4\,CuBr_2\ +\ 4\,LH +\ 4\,NaO_2CPh\ \overset{MeOH}{\rightarrow}\ [Cu_4Br_2(L)_4]Br_2\ +\ 4\,PhCO_2H\ +\ 4\,NaBr \tag{1}$$

Complex $[Ni_4(NO_3)_2(L)_4(H_2O)](NO_3)_2$ (**2**), crystallographically characterized as $\mathbf{2} \cdot 0.2H_2O \cdot 3EtOH$, was prepared by the 1:1 reaction between $Ni(NO_3)_2 \cdot 6H_2O$ and LH in $CH_2Cl_2$-EtOH, Equation (2), in a rather low yield (~30%). The use of $CH_2Cl_2$ was necessary to improve the quality of the obtained brown crystals. Use of $Et_3N$ in the reaction mixture gave the same complex in a powder form, Equation (3), but—somewhat to our surprise—with no significant yield improvement.

$$4\,Ni(NO_3)2\ 6H_2O\ +\ 4\,LH\ \overset{EtOH-CH_2Cl_2}{\rightarrow}\ [Ni_4(NO_3)_2(L)_4(H_2O)](NO_3)_2\ +4\,HNO_3\ +\ 23\,H_2O \tag{2}$$

$$4\,Ni(NO_3)2\ 6H_2O\ +\ 4\,LH\ +\ 4\,Et_3N\ \overset{EtOH-CH_2Cl_2}{\rightarrow}\ [Ni_4(NO_3)_2(L)_4(H_2O)](NO_3)_2\ +\ 4\,(Et_3NH)(NO_3)\ +\ 23\,H_2O \tag{3}$$

Depending on the Co(II): LH reaction ratio used, the $Co(ClO_4)_2 \cdot 6H_2O/LH$ reaction system gave two products in MeOH *under aerobic conditions*, namely $[Co_2(L)_3](ClO_4)_3$ (**3**), crystallographically formulated as $\mathbf{3} \cdot 0.8H_2O \cdot 1.3MeOH$, and $[Co(L)_2](ClO_4)$ (**4**) in moderate yields (~50%). Both products are Co(III) complexes, the atmospheric air oxygen being the oxidant; the oxidation is certainly facilitated by the N-rich environment from the ligand. The 2:3 reaction between $Co(ClO_4)_2 \cdot 6H_2O$ and LH gives complex **3** according to Equation (4). The addition of the base is not necessary for the formation and isolation of the dinuclear complex; its presence increases slightly the yield of the reaction. Use of an excess of LH (LH: $Co^{II}$ = 2:1) has provided access to the 1:2 mononuclear cationic complex 4; the use of base here is necessary for the isolation of the compound in satisfactory yields, Equation (5). Complex **3**

can be converted to compound **4** (albeit in a low yield) in MeOH under reflux, Equation (6). The yield can be impressively improved by the addition of base, e.g., LiOH, Equation (7).

$$4\,Co^{II}(ClO_4)_2 \cdot 6H_2O \; + \; 6\,LH \; + \; O_2 \xrightarrow{\text{MeOH}} 2\,[Co^{III,III}_2\,(L)_3](ClO_4)_3 \; + \; 2\,HClO_4 \; + 26\,H_2O \quad (4)$$

$$4\,Co^{II}(ClO_4)_2 \cdot 6H_2O \; + \; 8\,LH \; + 4\,Et_3N \; + \; O_2 \xrightarrow{\text{MeOH}} 4\,[Co^{III}(L)_2](ClO_4) \; + \; 4\,(Et_3NH)(ClO_4) \; + \; 26\,H_2O \quad (5)$$

$$[Co_2(L)_3](ClO_4)_3 \; + \; LH \xrightarrow[\text{T}]{\text{MeOH}} 2\,[Co\,(L)_2](ClO_4) \; + \; HClO_4 \quad (6)$$

$$[Co_2(L)_3](ClO_4)_3 \; + \; LH \; + \; LiOH \xrightarrow[\text{T}]{\text{MeOH}} 2\,[Co(L)_2](ClO_4) \; + \; LiClO_4 \; + \; H_2O \quad (7)$$

### 2.2. Spectroscopic Characterization in Brief

IR and ultraviolet/visible (UV/VIS) spectra of the complexes were obtained from analytically pure samples which have the formulae **1**, **2**, **3**·$H_2O$ and **4** (Section 3). In the IR spectrum of sample **3**·$H_2O$, the broad band centered at ~3420 $cm^{-1}$ is due to the $v$(OH) vibration of the lattice water [7]. The $v$(OH) vibration of coordinated $H_2O$ in the spectrum of **2** also appears in this region. The IR spectrum of the free ligand LH exhibits a medium-intensity band at 3316 $cm^{-1}$ and a very strong band at 1702 $cm^{-1}$, assigned to the $v$(NH) and $v$(C = O) vibrations, respectively [14,15]. The appearance of these stretching vibrations indicates that LH is present in its keto-amino form, and not in the enol-imino form drawn in Figure 1. Such vibrations are absent from the spectra of the complexes, the spectral regions 3400–3100 $cm^{-1}$ and 1700–1610 $cm^{-1}$ showing no bands. The absence of these bands indicates that (i) the ligands are deprotonated in the complexes and (ii) the carbon–oxygen bond of coordinated $L^-$ does not have an appreciable double bond character [23]; these facts are confirmed in the crystal structures of the complexes (*vide infra*). The highest wavenumber bands in the 2000–400 $cm^{-1}$ region are at 1594 (**1**), 1598 (**2**), 1606 (**3** $H_2O$) and 1602 (**4**) $cm^{-1}$, assigned to a pyridyl stretching vibration [7].

The KBr spectrum of **2** exhibits a strong sharp band at 1384 $cm^{-1}$, assigned to the $v_3(E')[v_d(NO)]$ vibrational mode of the planar ionic nitrate of $D_{3h}$ symmetry [28]. The absence of bands that would be indicative of the monodentate and bidentate coordinated nitrato groups (present in the structure of the cluster) is rather surprising. This suggests [29,30] that the nitrato ligands are replaced by bromides that are in excess in the KBr matrix, thus producing ionic nitrates ($KNO_3$); this replacement is facilitated by the pressure that is applied for the preparation of the KBr matrix. As expected, extra nitrato bands appear in the mull (nujol, hexachlorobutadiene) spectra of **2**. For example, the bands at 1501 and 1300 $cm^{-1}$ are assigned [28–30] to the $v_1(A_1)[v(N = O)]$ and $v_5(B_2)[v_{as}(NO_2)]$ vibrational modes, respectively, of the coordinated nitrato group. The separation of these two bands is large (~200 $cm^{-1}$), suggesting a bidentate nitrato ligand of $C_{2v}$ symmetry [28]. The $v_5(B_2)[v_{as}(NO_2)]$ band of the monodentate nitrato group could not be assigned with certainty in the mull spectra because other bands of stretching vibrations origin appear in the 1450–1350 $cm^{-1}$ region. The band at 1294 $cm^{-1}$ is a serious candidate for the $v_1(A_1)[v_s(NO_2)]$ vibration of the monodentate nitrato ligand which is expected around 1300 $cm^{-1}$ [28]. The spectra of **3**·$H_2O$ and **4** exhibit a strong band at 1090–1080 and a medium-intensity band at ~625 $cm^{-1}$, attributable to the IR-active $v_3(F_2)[v_d(Cl\text{-}O)]$ and $v_4(F_2)[\delta_d(OClO)]$ vibrations of the uncoordinated $T_d$ $ClO_4^-$ counterion, respectively [28].

The d-d spectrum of **1** in MeOH consists of a featureless band at 745 nm; this wavelength is fairly typical of a distorted square pyramidal or/and a tetragonally distorted six-coordinate geometry [31,32]. The spectrum also exhibits an absorption at ~370 nm assignable to a $Br^-$-to $Cu^{II}$ LMCT transition [32]. Copper(II) is relatively easy to reduce to copper(I) and the observed transition from a $\pi$ orbital of the bromo ligand to the singly occupied 3d orbital of $Cu^{II}$ occurs at a relatively low energy (~27,000 $cm^{-1}$). The d-d spectrum of **2** in MeOH consists of three bands at 365, 615 and 980 nm assignable [32,33] to the $^3A_{2g} \rightarrow {}^3T_{1g}$ (P), $^3A_{2g} \rightarrow {}^3T_{1g}$ (F) and $^3A_{2g} \rightarrow {}^3T_{2g}$ transitions, respectively, in an octahedral 3d$^8$ ligand field; the wavelengths are typical of Ni(II) chromophores possessing both N and O donors [32,33]. The UV/VIS spectra of concentrated solutions of **3**·$H_2O$ and **4** in MeCN are typical for low-spin octahedral

{Co$^{III}$N$_6$} and {Co$^{III}$N$_x$O$_{6-x}$} chromophores, respectively [32,34]. The low-spin octahedral ground term is $^1A_{1g}$ and there are two spin-allowed transitions, with lower lying spin triplet partners, all derived from $(t_{2g})^5(e_g)^1$. Under this scheme, the bands/shoulders at 395, 440, 580 and 735 nm in the spectrum of **3**·H$_2$O are assigned to the $^1A_{1g} \rightarrow {}^1T_{2g}$, $^1A_{1g} \rightarrow {}^1T_{1g}$, $^1A_{1g} \rightarrow {}^3T_{2g}$ and $^1A_{1g} \rightarrow {}^3T_{1g}$, respectively. The corresponding transitions in the spectrum of **4** appear at 405, 450, 590 and 760 nm.

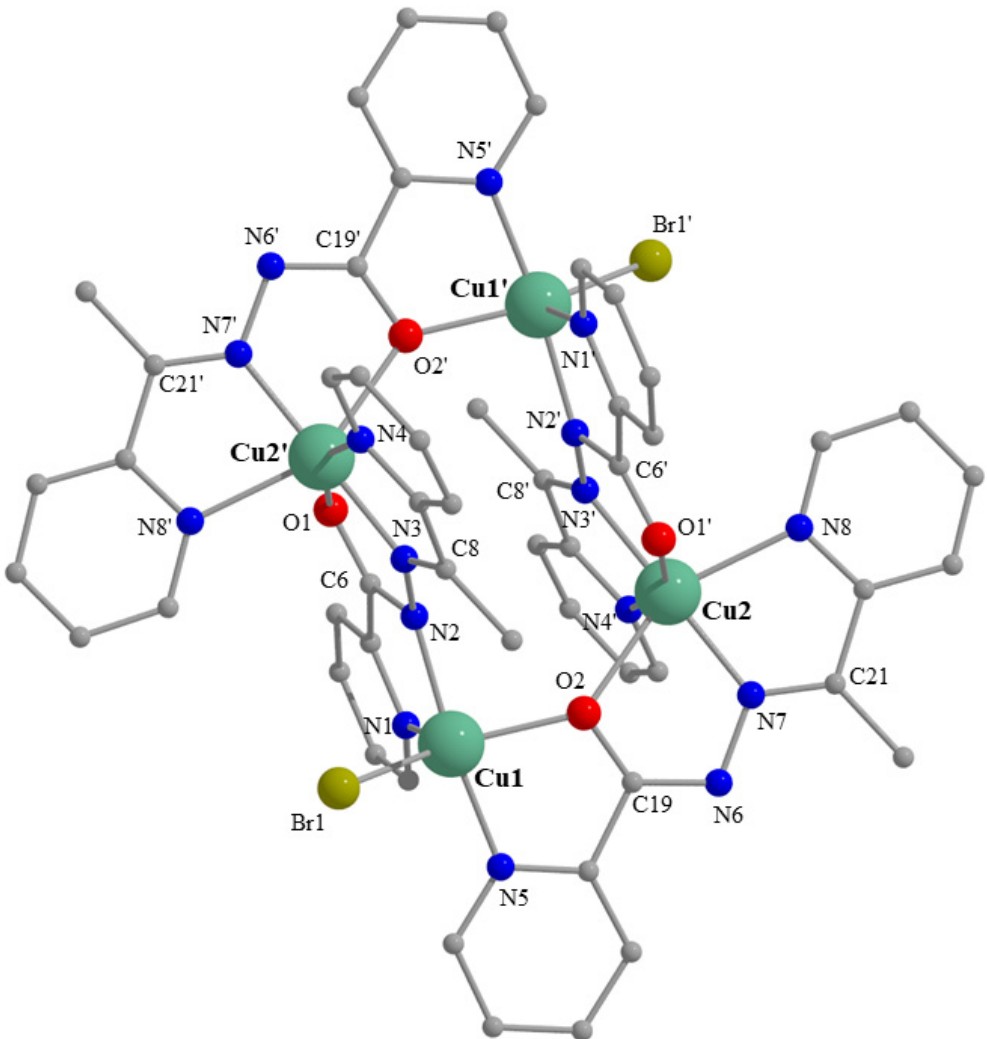

**Figure 1.** Partially labelled plot of the structure of the cation [Cu$_4$Br$_2$(L)$_4$]$^{+2}$ that is present in the crystal. Structure of **1**·0.8H$_2$O·MeOH. Symmetry code: (') = −$x$ + 1, −$y$ + 2, −$z$. A plot with thermal ellipsoids is presented in Figure S17.

The $^1$H nuclear magnetic resonance (NMR) spectra of the diamagnetic, analytically pure samples **3**·H$_2$O and **4** in deuterated dimethyl sulfoxide (DMSO-d$_6$) are almost identical, except the extra peak at $\delta$ 3.17 ppm in the former due to the protons of the lattice H$_2$O. Singlet resonances of the methyl protons were observed at the rather low-field $\delta$ value of 3.35 ppm, while the resonances of the eight pyridyl rings appear in the $\delta$ region 8.60–7.45 ppm; the integration ratio is 3:8, as expected. These facts indicate the presence of a single L$^-$ species in solution, but our data—combined with literature reports [18]—do not permit safe conclusions concerning the coordinated or non-coordinated nature of the anionic species in solution.

Representative spectra of the complexes are presented in Figures S1–S16.

## 2.3. Description of Structures

　　The structures of the four complexes have been solved by single-crystal, X-ray crystallography. Aspects of the molecular and crystal structures are shown in Figures 1–8 and Figures S17–S21. Crystallographic data are presented in Table S1, while numerical data concerning interatomic distances, bond angles and H-bonding interactions are listed in Tables 1–4 and Figures S2–S5.

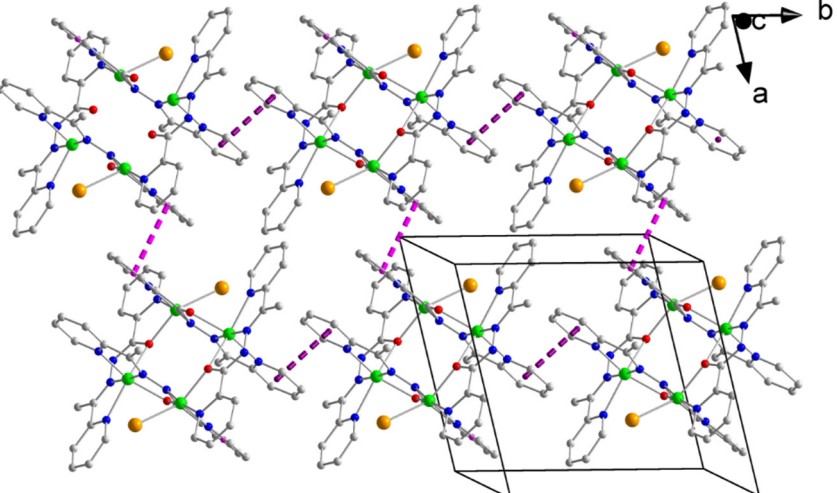

**Figure 2.** A layer of the tetranuclear cations of complex **1**·0.8H$_2$O·MeOH parallel to the (001) plane. The dashed pink and violet lines indicate π–π stacking interactions between centrosymmetrically-related pairs of aromatic rings containing the N1 and N4 atoms, respectively.

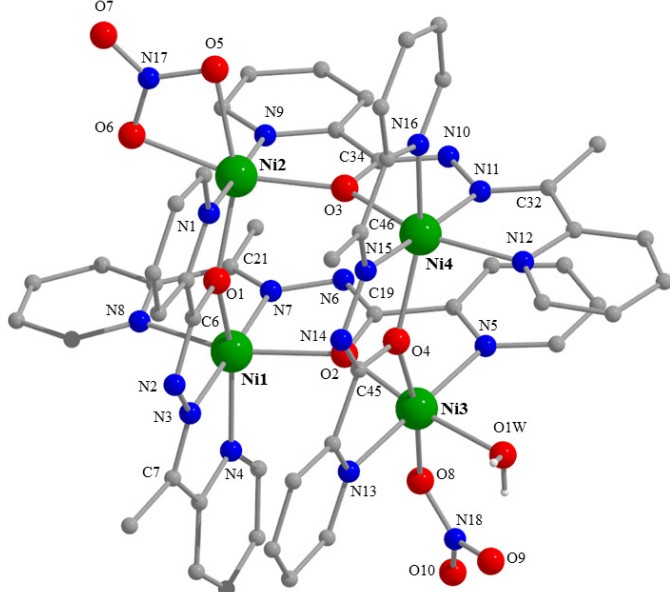

**Figure 3.** Partially labelled plot of the structure of the cation [Ni$_4$(NO$_3$)$_2$(L)$_4$(H$_2$O)]$^{2+}$ that is present in the.crystal structure of **2**·0.2H$_2$O·3EtOH. A plot with thermal ellipsoids is presented in Figure S19.

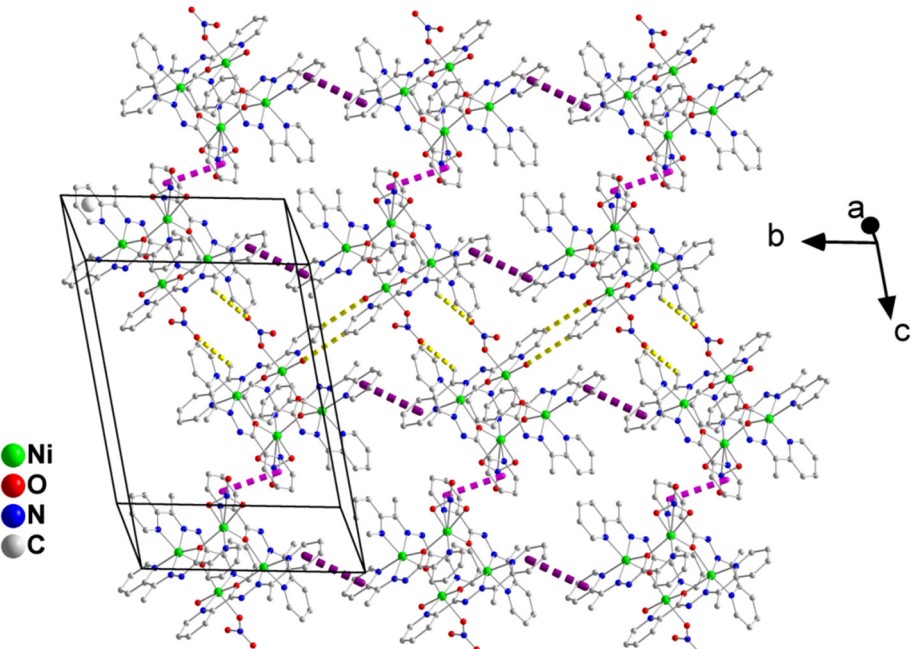

**Figure 4.** A layer of the tetranuclear cations of complex **2**·0.2H$_2$O·3EtOH parallel to the (100) plane. The dashed pink and violet lines indicate interchain and intrachain, respectively, π–π stacking interactions. The dashed yellow lines represent interstripe H-bonding interactions.

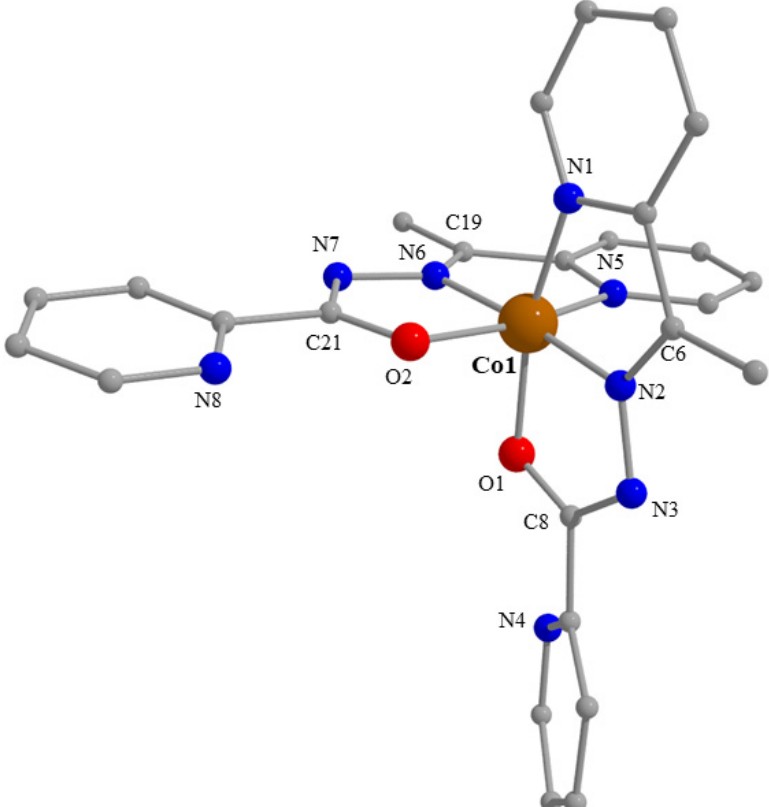

**Figure 5.** Partially labelled plot of the structure of the cation [Co(L)$_2$]$^+$ that is present in the crystal structure of **4**. A plot with thermal ellipsoids is presented in Figure S20.

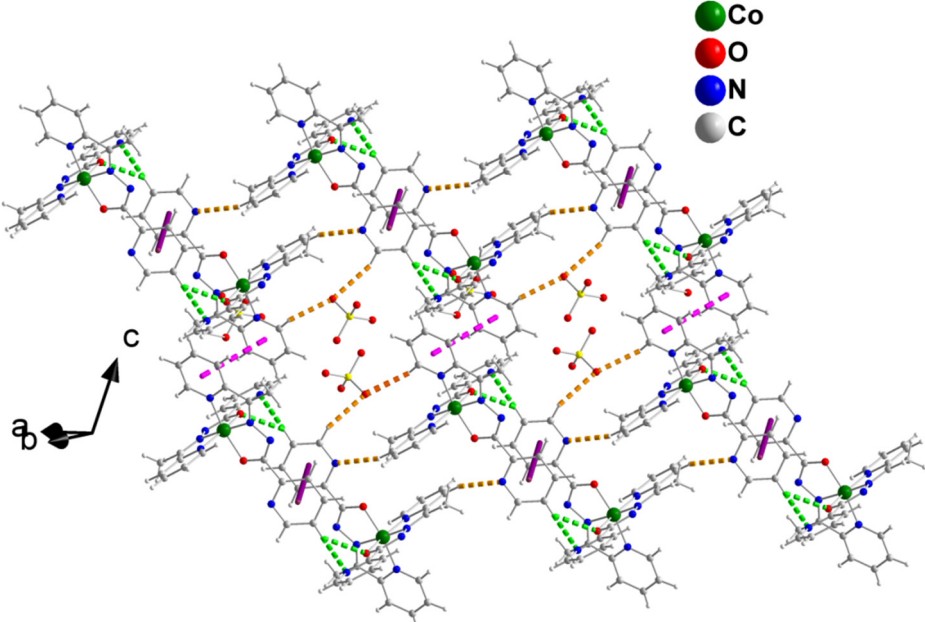

**Figure 6.** A layer of **4** parallel to the (1–10) plane. The dashed pink and solid violet lines indicate π–π. interactions between pairs of N1- and N4-containing rings, respectively. The dashed green lines represent the C12-H(C12)$\cdots$N8 and C12-H(C12)$\cdots$O2 intrachain H bonds. The dashed orange lines represent the C17-H(C17)$\cdots$N4, C1-H(C1)$\cdots$O5 and C13-H(C13)$\cdots$O5 interchain H bonds. Atoms C1, C12, C13 and C17 are aromatic carbon atoms not labelled in Figure 5 and Figure S4. Atom O5 belongs to the $ClO_4^-$ counterion not shown in Figure 5 and Figure S4.

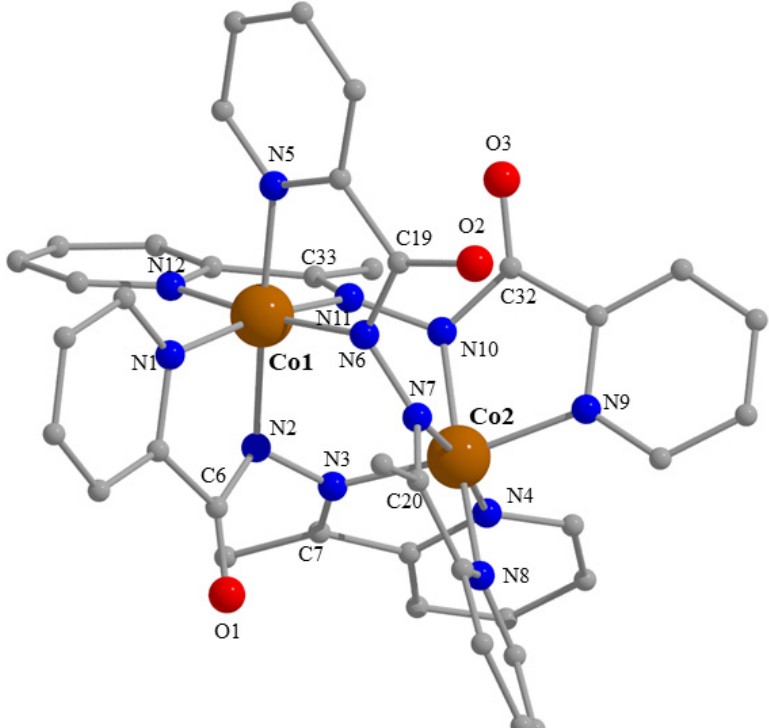

**Figure 7.** Partially labelled plot of the cation $[Co_2(L)_3]^{3+}$ that is present in the crystal structure of **3**·0.8$H_2O$·1.3MeOH. A plot with thermal ellipsoids is presented in Figure S21 (left).

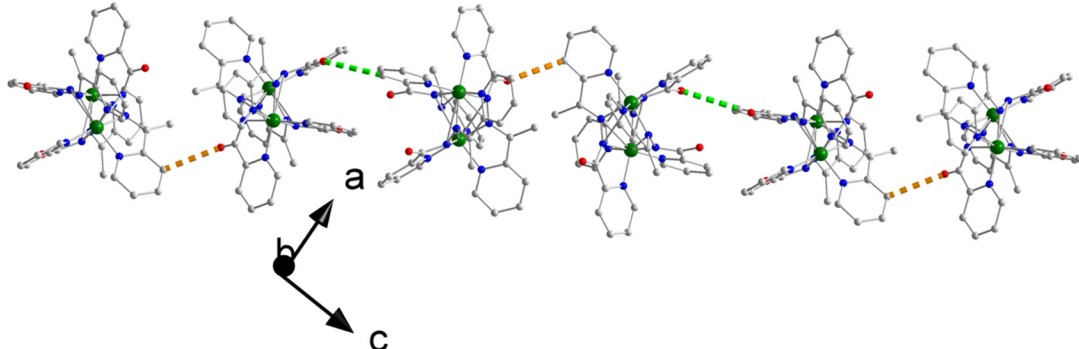

**Figure 8.** A chain of $[Co_2(L)_3]^{3+}$ cations parallel to the [101] direction in the crystal structure of **3**·0.8$H_2O$·1.3MeOH. The dashed orange and green lines indicate the H-bonding interactions C9-H(C9)···O1 and C17-H(C17)···O3, respectively. Atoms C9 and C17 are aromatic carbon atoms, not labelled in Figure 7 and Figure S21.

**Table 1.** Selected interatomic distances (Å) and bond angles (°) for complex **1**·0.8$H_2O$·MeOH [a].

| Interatomic Distances (Å) | | Bond Angles (°) | |
|---|---|---|---|
| Cu1-N1 | 2.143(5) | N2-Cu1-N5 | 172.7(2) |
| Cu1-N2 | 1.989(5) | O2-Cu1-Br1 | 147.7(1) |
| Cu1-O2 | 2.073(4) | N1-Cu1-Br1 | 117.0(1) |
| Cu1-N5 | 1.994(5) | N1-Cu1-N2 | 80.3(2) |
| Cu1-Br1 | 2.450(1) | N1-Cu1-N5 | 97.0(2) |
| Cu2-N7 | 1.980(5) | N1-Cu1-O2 | 95.3(2) |
| Cu2-N8 | 2.193(5) | N2-Cu1-O2 | 93.4(2) |
| Cu2-O2 | 2.331(4) | N5-Cu1-Br1 | 95.1(1) |
| Cu2-N3′ | 1.975(5) | N3′-Cu2-N7 | 175.5(2) |
| Cu2-N4′ | 2.054(5) | N8-Cu2-O2 | 148.2(2) |
| Cu2-O1′ | 2.117(4) | N4′-Cu2-O1′ | 155.4(2) |
| C6-O1 | 1.257(7) | N3′-Cu2-N4′ | 79.1(2) |
| C6-N2 | 1.346(8) | N7-Cu2-N8 | 77.4(2) |
| N2-N3 | 1.384(7) | O2-C19-N6 | 126.8(5) |
| N3-C8 | 1.290(8) | C19-N6-N7 | 110.3(5) |
| C19-O2 | 1.287(7) | N6-N7-C21 | 118.1(5) |
| C19-N6 | 1.331(7) | O1-C6-N2 | 125.0(5) |
| N6-N7 | 1.401(6) | C6-N2-N3 | 110.2(5) |
| N7-C21 | 1.281(7) | N2-N3-C8 | 123.7(5) |

[a] Symmetry code (′) = $-x + 1$, $-y + 2$, $-z$.

The crystal structure of **1**·0.8$H_2O$·MeOH consists of tetranuclear cations $[Cu_4Br_2(L)_4]^{+2}$ (Figure 1 and Figure S17), Br$^-$ counterions, and lattice $H_2O$ and MeOH molecules. The cation possesses a crystallographic inversion center in the midpoint of the Cu1···Cu1′ (or Cu2···Cu2′) distance. The cation is a rectangular [2 × 2] grid; all the metal centers are strictly coplanar (by symmetry). The sides of the rectangle are 4.121(1) Å (Cu1···Cu2/Cu1′···Cu2′) and 4.797(1) Å (Cu1···Cu2′/Cu1′···Cu2), and the diagonals are 6.690(1) Å (Cu1···Cu1′) and 5.935(1) Å (Cu2···Cu2′).

**Table 2.** Selected interatomic distances (Å) and bond angles (°) for complex **2**·0.2H$_2$O·3EtOH.

| Interatomic Distances (Å) | | Bond Angles (°) | |
|---|---|---|---|
| Ni1···Ni2 | 3.922(1) | O1-Ni1-N4 | 154.1(1) |
| Ni1···N3 | 3.938(1) | O2-Ni1-N8 | 153.9(1) |
| Ni1···N4 | 5.526(1) | N3-Ni1-N7 | 177.9(1) |
| Ni2···Ni3 | 5.559(1) | O1-Ni1-O2 | 91.4(1) |
| Ni2···Ni4 | 3.921(1) | O2-Ni1-N3 | 105.3(1) |
| Ni3···Ni4 | 3.895(1) | N3-Ni1-N4 | 78.0(1) |
| Ni1-O1 | 2.147(2) | O1-Ni2-O5 | 158.2(1) |
| Ni1-O2 | 2.144(2) | O3-Ni2-O6 | 165.1(1) |
| Ni1-N3 | 1.988(2) | N1-Ni2-N9 | 177.5(1) |
| Ni1-N4 | 2.083(2) | O1-Ni2-O3 | 95.0(1) |
| Ni1-N7 | 1.984(2) | O5-Ni2-O6 | 60.7(1) |
| Ni1-N8 | 2.107(2) | O6-Ni2-N9 | 93.3(1) |
| Ni2-O1 | 2.049(2) | O2-Ni3-O1W | 169.9(1) |
| Ni2-O3 | 2.054(2) | O4-Ni3-O8 | 171.1(1) |
| Ni2-O5 | 2.161(2) | N5-Ni3-N13 | 179.1(1) |
| Ni2-O6 | 2.113(2) | O2-Ni3-O4 | 90.5(1) |
| Ni2-N1 | 2.041(2) | O1W-Ni3-O8 | 93.9(1) |
| Ni2-N9 | 2.036(2) | O8-Ni3-N5 | 86.8(1) |
| Ni3-O2 | 2.068(2) | O3-Ni4-N12 | 154.5(1) |
| Ni3-O4 | 2.052(2) | O4-Ni4-N16 | 154.7(1) |
| Ni3-O8 | 2.121(2) | N11-Ni4-N15 | 174.5(1) |
| Ni3-O1W | 2.080(2) | O3-Ni4-O4 | 94.4(1) |
| Ni3-N5 | 2.048(2) | N11-Ni4-N12 | 78.5(1) |
| Ni3-N13 | 2.055(2) | N15-Ni4-N16 | 78.4(1) |
| Ni4-O3 | 2.144(2) | O8-N18-O9 | 120.7(3) |
| Ni4-O4 | 2.126(2) | O8-N18-O10 | 118.8(3) |
| Ni4-N11 | 1.987(2) | O9-N18-O10 | 120.5(3) |
| Ni4-N12 | 2.093(2) | O5-N17-O6 | 116.2(3) |
| Ni4-N15 | 1.979(2) | O5-N17-O7 | 122.2(4) |
| Ni4-N16 | 2.091(2) | O6-N17-O7 | 121.6(4) |

**Table 3.** Selected bond lengths (Å) and angles for complex **4**.

| Bond Lengths (Å) [a] | | Bond Angles (°) [a] | |
|---|---|---|---|
| Co1-O1 | 1.899(2) | O1-Co1-N1 | 165.2(1) |
| Co1-N1 | 1.927(2) | O2-Co1-N5 | 165.4(1) |
| Co1-N2 | 1.855(2) | N2-Co1-N6 | 174.7(1) |
| Co1-O2 | 1.892(2) | O1-Co1-N2 | 82.5(1) |
| Co1-N5 | 1.915(2) | O1-Co1-O2 | 91.3(1) |
| Co1-N6 | 1.857(2) | O2-Co1-N6 | 82.6(1) |
| C8-O1 | 1.290(3) | N2-Co1-N5 | 100.7(1) |
| C8-N3 | 1.329(3) | O1-C8-N3 | 125.1(2) |
| N3-N2 | 1.391(3) | C8-N3-N2 | 106.3(2) |
| N2-C6 | 1.298(3) | N3-N2-C6 | 123.7(2) |
| C21-O2 | 1.306(3) | O2-C21-N7 | 124.2(2) |
| C21-N7 | 1.317(3) | C21-N7-N6 | 107.3(2) |
| N7-N6 | 1.383(3) | N7-N6-C19 | 123.9(2) |
| N6-C19 | 1.299(3) | O3-Cl1-O4 | 109.7(1) |
| Cl1-O3 | 1.440(2) | O3-Cl1-O6 | 110.7(1) |
| Cl1-O4 | 1.435(2) | O4-Cl1-O5 | 108.8(1) |
| Cl1-O5 | 1.442(2) | O5-Cl1-O6 | 108.8(1) |
| Cl1-O6 | 1.428(2) | | |

[a] The Cl-O bond lengths and O-Cl-O bond angles refer to the perchlorate counter ion not shown in Figure 5 and Figure S20.

**Table 4.** Selected interatomic distances (Å) and bond angles (°) for complex **3**·0.8 H$_2$O·1.3MeOH.

| Interatomic Distances (Å) | | Bond Angles (°) | |
|---|---|---|---|
| Co1···Co2 | 3.467(2) | N1-Co1-N11 | 170.6(3) |
| Co1-N1 | 1.953(6) | N2-Co1-N5 | 172.5(3) |
| Co1-N2 | 1.906(6) | N6-Co1-N12 | 172.0(3) |
| Co1-N5 | 1.951(6) | N1-Co1-N2 | 81.3(3) |
| Co1-N6 | 1.899(6) | N2-Co1-N6 | 92.0(3) |
| Co1-N11 | 1.910(6) | N5-Co1-N6 | 81.6(3) |
| Co1-N12 | 1.931(6) | N11-Co1-N12 | 82.1(3) |
| Co2-N3 | 1.900(6) | N3-Co2-N9 | 170.3(3) |
| Co2-N4 | 1.935(6) | N4-Co2-N7 | 171.1(2) |
| Co2-N7 | 1.901(6) | N8-Co2-N10 | 171.4(2) |
| Co2-N8 | 1.946(6) | N3-Co2-N4 | 81.7(2) |
| Co2-N9 | 1.942(7) | N4-Co2-N8 | 93.1(2) |
| Co2-N10 | 1.909(6) | N7-Co2-N8 | 81.6(3) |
| C6-O1 | 1.242(9) | N8-Co2-N9 | 94.1(3) |
| C6-N2 | 1.363(9) | O1-C6-N2 | 126.9(7) |
| N2-N3 | 1.407(7) | C6-N2-N3 | 114.6(6) |
| N3-C7 | 1.310(8) | N2-N3-C7 | 121.6(6) |
| C19-O2 | 1.319(9) | O2-C19-N6 | 126.1(8) |
| C19-N6 | 1.311(8) | C19-N6-N7 | 119.4(6) |
| N6-N7 | 1.384(8) | N6-N7-C20 | 120.5(6) |
| N7-C20 | 1.331(9) | O3-C32-N10 | 124.9(9) |
| C32-O3 | 1.253(9) | C32-N10-N11 | 117.9(6) |
| C32-N10 | 1.348(8) | N10-N11-C33 | 121.3(6) |
| N10-N11 | 1.400(8) | | |
| N11-C33 | 1.292(8) | | |

　　　Two 2.21011 (Harris notation [35]) anionic L$^-$ ligands bridge adjacent Cu$^{II}$ atoms on the short sides of the rectangle through their alkoxide oxygen atoms (O2, O2'). Two 2.11111 L$^-$ ligands bridge adjacent Cu$^{II}$ atoms on the long sides of the rectangle with their diazine groups (N2-N3, N2'-N3'). The two different coordination modes are shown in Scheme 2. The Cu2/Cu2' ions are 6-coordinate with a distorted octahedral geometry, the *trans* coordination angles being in the range 148.2(2)–175.5(2)°. The Jahn–Teller axis is defined as N8-Cu2-O2 [Cu2-N8 = 2.193(5) Å, Cu2-O2 = 2.331(4) Å]. The Cu1/Cu1' ions are 5-coordinate with a {Cu$^{II}$N$_3$OBr} chromophore; the access to the sixth donor site is blocked by the presence of the methyl groups. The coordination geometry can be described as either distorted square pyramidal or distorted trigonal bipyramidal. Analysis of the shape-determining angles using the approach of Addison and Reedjik [36] yields a value for the trigonality index, $\tau$, of 0.42 ($\tau = 0$ and 1 for square pyramidal and trigonal bipyramidal geometry, respectively). Adopting the square pyramidal description, the basal plane consists of donor atoms N2, N5, O2 and Br1, with atom N1 occupying the apical position. The alternative distorted trigonal bipyramidal description places donor atoms N2 and N5 at the axial positions, and atoms N1, O2 and Br1 at the equatorial sites. The Cu1-O2-Cu2 (Cu1'-O2'-Cu2') angle is 138.6(2)°. Establishing the site of deprotonation is sometimes difficult for L$^-$ and related ligands. The C6-O1, C6-N2, N2-N3, N3-C8 and C19-O2, C19-N6, N6-N7, N7-C21 bond distances (Table 1) indicate a charge delocalization within the OCNNC backbone of the two crystallographically independent, deprotonated ligands of the tetranuclear cation. If we should define the deprotonated atom, the relatively long C6-O1 and C19-O2 distances [1.257(7) and 1.287(7) Å, respectively] suggest that the O atoms are the principal sites of deprotonation [14].

**Scheme 2.** The to date crystallographically established coordination modes of LH and L⁻, and the Harris notation that describes these modes. The neutral ligand (LH) is shown in the keto-amino form. The central OCNNC unit of the anionic ligand has been drawn in a manner that emphasizes its delocalized description that appears in most complexes.

The tetranuclear cations of the complex form layers parallel to the (001) plane through π–π stacking interactions (Figure 2). The Cg1···Cg1″ and Cg2···Cg2‴ distances are 3.713(1) and 3.661(1) Å, respectively, where Cg1 is the centroid of the N1-containing aromatic ring and Cg2 is the centroid of

the N4-containing ring [symmetry codes: ('') = −$x$ + 2, −$y$ + 2, −$z$; (''') = −$x$ + 1, −$y$ + 1, −$z$]. The cations that form the layers interact further through π–π overlaps between centrosymmetrically-related ligands involving the N5, N8- and N5\*, N8\*-containing rings which are at a 3.34 Å distance and belong to adjacent layers (Figure S2), thus building the 3D architecture of the structure [symmetry code: (\*) = −$x$ + 1, −$y$ + 2, −$z$ + 1]. The Br$^-$ counterions, and the solvate $H_2O$ and MeOH molecules are hosted in the lattice through H bonds (Table S18).

The crystal structure of **2**·0.2$H_2O$·3EtOH consists of tetranuclear cations [Ni$_4$(NO$_3$)$_2$(L)$_4$($H_2O$)]$^{2+}$ (Figure 3 and Figure S19), NO$_3^-$ counterions, as well as EtOH and $H_2O$ molecules. The cation is a square [2 × 2] grid. The four metal ions are practically coplanar, the distances from their best mean plane being: Ni1 0.0016(4) Å, Ni2 0.0016(4) Å, Ni3 0.016(4) Å and Ni4 0.017(4) Å. The sides of the square are in the 3.895(1)–3.938(1) Å range, and the diagonals are 5.526(1) Å (Ni1···Ni4) and 5.559(1) Å (Ni2···Ni3).

The two Ni$^{II}$ atoms of each side of the square are bridged by the alkoxide O atom (O1, O2, O3, O4) of one 2.21011 L$^-$ ligand (Scheme 2) and the core of the cluster cation is thus {Ni$_4$(OR)$_4$}$^{4+}$. One terminal $H_2O$ ligand and one monodentate nitrato group are coordinated to Ni3, while the two vacant coordination sites at Ni2 are occupied by two oxygen atoms (O5, O6) of a bidentate chelating nitrato group. The metal ions are all six-coordinate with distorted octahedral geometries, and the chromophores are {Ni1N$_4$O$_2$}, {Ni2N$_2$O$_4$}, {Ni3N$_2$O$_4$} and {Ni4N$_4$O$_2$}. The Ni-O and Ni-N bond lengths are typical [7,37,38] for octahedral Ni(II) complexes. The C-O$_{bridging}$ bond distances are in the narrow 1.295(3)–1.306(3) Å, indicating the predominance of single CO bond character [15]. The Ni-O-Ni angles are ~138°. Within the cation, there is a strong H bond with O1W as donor and the non-coordinate O9 atom of the nitrato ligand as acceptor (Figure 3, Table S3).

The tetranuclear cations of the complex form layers parallel to the (100) plane through π–π stacking interactions and H bonds (Figure 4). The π–π overlap of the N8- and N12′-containing rings of neighboring cluster cations create chains parallel to the b axis [symmetry code: (′) = −$x$, $y$ − 1, $z$]; the distance between their centroids is 4.076(1) Å and the angle between the ring planes is 9.2(2)°. Through the π–π overlap of centrosymmetrically-related rings that contain N1 and N1″ [symmetry code: ('') = −$x$ + 2, −$y$ − 1, −$z$ − 1; the distance between the planes is 3.36(2) Å] and belong to neighboring cluster cations, pairs of chains are formed extending parallel to the b axis and thus double chains are created. Cations belonging to neighboring double chains interact through C20-H$_C$(C20)···O10 and C15-H(C15)···O1W H bonds (Table S3), where C20 and C15 are methyl and aromatic carbon atoms, respectively, forming layers of double chains parallel to the (100) plane. These layers are stacked along the a axis. Nitrate counterions and lattice EtOH molecules are hosted between the layers through an extensive H-bonding network (Table S3); these species are linked with each other and also connect neighboring layers building the 3D architecture of the structure. The donors of these H bonds are mainly the oxygen atoms of the lattice $H_2O$ and EtOH molecules, while the acceptors are counter nitrate and lattice EtOH oxygen atoms.

We start the structural descriptions of the Co(III) complexes with the simplest compound, i.e., **4**. The crystal structure of **4** consists of ions [Co(L)$_2$]$^+$ (Figure 5 and Figure S20) and ClO$_4^-$ in an 1:1 ratio. Both mutually perpendicular L$^-$ ions act as tridentate chelating, meridional 1.10011 ligands (Scheme 2), each ligand forming two practically planar 5-membered chelating rings. The metal ion is coordinated by two alkoxide oxygen atoms (O1, O2), two 2-pyridyl nitrogen atoms (N1, N5) and two hydrazone nitrogen atoms (N2, N6) resulting in a distorted octahedral geometry. The *trans* angles of the octahedron are 165.2(1), 165.4(1) and 174.7(1)°. Due to the meridional character of each anionic ligand, two pairs of *trans* coordination sites are each occupied by donor atoms of the same L$^-$ group (O1/N1, O2/N5), whereas the third pair of coordination sites consists of nitrogen atoms (N2, N6) from different ligands. The Co-(O, N) bond lengths are in the range 1.855(2)–1.927(2) Å and agree very well with values observed for low-spin Co(III) ions in octahedral environments [35,39,40]. The two Co-N bond lengths for each ligand differ [1.927(2) vs. 1.855(2) Å, and 1.915(2) vs. 1.857(2) Å]; the shorter

bond distance (stronger bond) pertains to the hydrazone nitrogen atoms (N2, N6), probably due to the presence of some negative charge on these atoms (because of delocalization).

The mononuclear cations of **4** form chains parallel to the [111] crystallographic direction through π–π stacking interactions between centrosymmetrically-related N(1)- and N(1′)-containing rings [symmetry code: (′) = −$x$, −$y$ + 1, −$z$ + 1; the distance between the planes is 3.62(1) Å] and between centrosymmetrically-related N(4)- and N(4′)-containing rings [symmetry code: (′) = −$x$ + 1, −$y$ + 2, −$z$ + 2; the distance between the planes is 3.36(1) Å] (Figure 6). The cations of a given chain are further linked through C12-H(C12)···N8 and C12-H(C12)···O2 bifurcated H bonds (Table S4); C12 is an aromatic carbon atom. The chains are connected through C17-H(C17)···N4, C1-H(C1)···O5 and C13-H(C13)···O5 H bonds forming layers parallel to the (1–10) plane; atoms C1, C13 and C17 are aromatic carbon atoms and O5 belongs to the $ClO_4^-$ counterion. These layers are stacked along the b axis and linked through C14-H(C14)···O4, C13-H(C13)···O5 and C7-H$_B$(C7)···N7 H bonds (Table S4); C14 is an aromatic carbon atom and C7 is a methyl carbon atom.

The crystal structure of **3**·0.8H$_2$O·1.3MeOH consists of dinuclear cations $[Co_2(L)_3]^{3+}$ (Figure 7 and Figure S21) and $ClO_4^-$ counterions in an 1:3 ratio, as well as solvent $H_2O$ and MeOH lattice molecules. The cation has approximate D$_3$ symmetry with the three bis(bidentate) 2.01111 L$^-$ ligands (Scheme 2) wrapped around the Co1···Co2 axis in such a way as to give each metal ion a distorted octahedral {Co$^{III}$N$_6$} coordination. The Co$^{III}$-N bond lengths are in the range 1.899(6)–1.953(6) Å confirming [34,40] the low-spin character of the two 3d$^6$ metal ions. These bond distances follow the same trend seen for **4**, i.e., Co$^{III}$-N (hydrazone) < Co$^{III}$-N (2-pyridyl). The *trans* coordination angles for Co1 and Co2 are in the ranges 170.6(3)–172.5(3)° and 170.3(3)–171.4(2)°, respectively. Each six-coordinate Co$^{III}$ atom adopts a *fac* configuration; the 2-pyridyl nitrogen atoms (and also the hydrazone nitrogen atoms) are arranged so as to define one face of the octahedron for each metal center. The Co1···Co2 distance is rather short, i.e., 3.467(2) Å, due to the presence of three diatomic bridges between the two metal ions. The cation can be considered as a *pseudo* triple helicate. We prefer the term *pseudo* (or helicate-type) because ligands that form helicates generally comprise two bidentate coordinating groups A-B connected via some linker [41–46]; this linker is absent in L$^-$. In most cases (including our complex) the helicities of the two metal centers (i.e., Δ and Λ) may be mechanically coupled, so for example formation of a Δ configuration at the first metal ion induces Δ configuration at the second; this is clearly illustrated in Figure S21c for one of the $[Co_2(L)_3]^{3+}$ cations of compound **3**·0.8H$_2$O·1.3MeOH which is a homochiral Δ,Δ ("right-handed") system. Due to the centrosymmetric space group ($P\bar{1}$), both enantiomers (Δ,Δ and Λ,Λ) are present in the helical structure of the complex [41,43], i.e., the crystal is a racemic sample.

The dinuclear cations of **3**·0.8H$_2$O·1.3MeOH form chains parallel to the [101] crystallographic direction through C9-H(C9)···O1 and C17-H(C17)···O3 H bonds (Figure 8, Table S5). These chains build the 3D architecture of the complex through H bonds involving aromatic carbon atoms as donors, and $ClO_4^-$ and lattice $H_2O$ oxygen atoms as acceptors (Table S5).

Complexes **1**–**4** join a family of structurally characterized coordination complexes containing LH and L$^-$ as ligands. The to date characterized metal complexes are listed in Table 5 along with information about the coordination mode of the ligands, the nuclearity/dimensionality of the products and the coordination geometry of the metal ions involved. Perusal of Table 5 shows that **2**, **3** and **4** are the first structurally characterized nickel and cobalt complexes of any form (neutral or anionic) of the ligand. The L$^-$ coordination modes 2.01111 and 1.10011 (Scheme 2) have been crystallographically established for the first time in the Co(III) complexes **3** and **4**. The rectangular [2 × 2] grid complex $[Cu_4Br_2(L)_4]Br_2$ (**1**) is structurally similar to complex $[Cu_4(L)_4(H_2O)_2](NO_3)_4$ [14], albeit with slight differences resulting from the different nature of the ancillary inorganic ligand (Br$^-$ vs. $H_2O$). Many (but not all) structural characteristics of the square [2 × 2] grid complex $[Ni_4(NO_3)_2(L)_4H_2O](NO_3)_2$ (**2**) resemble those of the Mn(II) complexes $[Mn_4(CF_3SO_3)(L)_4(H_2O)_3](CF_3SO_3)_3$ [15] and $[Mn_4(N_3)_4(L)_4]$ [19].

**Table 5.** To date crystallographically characterized metal complexes of LH and L⁻, and relevant structural information.

| Complex [a] | Coordination Mode [b,c] | Nuclearity/Dimensionality | Coordination Geometry | Ref. |
|---|---|---|---|---|
| $[CdBr_2(LH)]$ | $\eta^1{:}\eta^1{:}\eta^1(1.10011)$ | Mononuclear | Trigonal bipyramidal | [16] |
| $[Ln(NO_3)_3(LH)(MeOH)_2]$ (Ln = La, Ce) | $\eta^1{:}\eta^1{:}\eta^1(1.10011)$ | Mononuclear | Capped pentagonal antiprismatic | [17] |
| $[Nd(NO_3)_3(LH)(H_2O)]$ | $\eta^1{:}\eta^1{:}\eta^1(1.10011)$ | Mononuclear | Bicapped square antiprismatic | [17] |
| $[PdCl_2(LH)]$ | $\eta^1{:}\eta^1(1.01100)$ | Mononuclear | Square planar | [18] |
| $[HgX_2(LH)]$ (X = Cl, Br) | $\eta^1{:}\eta^1{:}\eta^1(1.10011)$ | Mononuclear | Square pyramidal | [20] |
| $[HgI_2(LH)(H_2O)]$ | $\eta^1{:}\eta^1{:}\eta^1(1.10011)$ | Mononuclear | Octahedral | [20] |
| $\{[Pb_3Br_6(LH)_2]\}_n$ | $\eta^1{:}\eta^1{:}\eta^1(1.10011)$ | 1D metal-organic ribbon | 7-coordinate [d], Octahedral | [21] |
| $[PdCl(L)]$ | $\eta^1{:}\eta^1{:}\eta^1(1.01011)$ | Mononuclear | Square planar | [18] |
| $[PdCl(L)]$ | $\eta^1{:}\eta^1{:}\eta^1(1.01101)$ | Mononuclear | Square planar | [18] |
| $[Cu_4(L)_4(H_2O)_2](NO_3)_4$ | $\eta^1{:}\eta^2{:}\eta^1{:}\eta^1{:}\mu_2(2.21011), \eta^1{:}\eta^1{:}\eta^1{:}\eta^1{:}\eta^1{:}\mu_2(2.11111)$ | Rectangular [2 × 2] grid | Square pyramidal, Octahedral | [14] |
| $[Mn_4(CF_3SO_3)(L)_4(H_2O)_3](CF_3SO_3)_3$ | $\eta^1{:}\eta^2{:}\eta^1{:}\eta^1{:}\mu_2(2.21011)$ | Square [2 × 2] grid | Octahedral | [15] |
| $[Mn_5(L)_6](ClO_4)_4$ | $\eta^1{:}\eta^2{:}\eta^1{:}\eta^1{:}\mu_2(2.21011)$ | Trigonal bipyramidal $\{Mn_5(\mu_2\text{-}OR)_6\}^{4+}$ core | Octahedral | [15] |
| $[Mn_4(N_3)_4(L)_4]$ | $\eta^1{:}\eta^2{:}\eta^1{:}\eta^1{:}\mu_2(2.21011)$ | Square [2 × 2] grid | Octahedral | [19] |
| $[Cu_4Br_2(L)_4]Br_2$ (1) | $\eta^1{:}\eta^2{:}\eta^1{:}\eta^1{:}\mu_2(2.21011), \eta^1{:}\eta^1{:}\eta^1{:}\eta^1{:}\eta^1{:}\mu_2(2.11111)$ | Rectangular [2 × 2] grid | Square pyramidal, Octahedral | this work |
| $[Ni_4(NO_3)_2(L)_4(H_2O)](NO_3)_2$ (2) | $\eta^1{:}\eta^2{:}\eta^1{:}\eta^1{:}\mu_2(2.21011)$ | Square [2 × 2] grid | Octahedral | this work |
| $[Co_2(L)_3](ClO_4)_3$ (3) | $\eta^1{:}\eta^1{:}\eta^1{:}\eta^1{:}\mu_2(2.01111)$ | Dinuclear helicate | Octahedral | this work |
| $[Co(L)_2](ClO_4)$ (4) | $\eta^1{:}\eta^1{:}\eta^1(1.10011)$ | Mononuclear | Octahedral | this work |

[a] Lattice solvent molecules have been omitted. [b] Both the η/μ convention and the Harris notation are provided. [c] See Scheme 2. [d] The coordination geometry of the 7-coordinate Pb^II. Atom was not mentioned.

## 2.4. Magnetochemistry

Direct current (dc) magnetic susceptibility data ($\chi$) on dried polycrystalline, analytically pure samples of **1** and **2** were collected in the 1.8–310 K range in an applied field of 1 kOe. The data are plotted as $\chi$ vs. *T* and $\chi T$ vs. *T* plots in Figure 9; Figure 10. Magnetization data were collected as a function of field from 0 to 50 kOe at 1.8 K and are presented in Figures S22 and S23. Although *M* vs. *H* per mole may be presented in units of Bohr magneton, we present the data in units of emu/mol because current literature uses these units commonly.

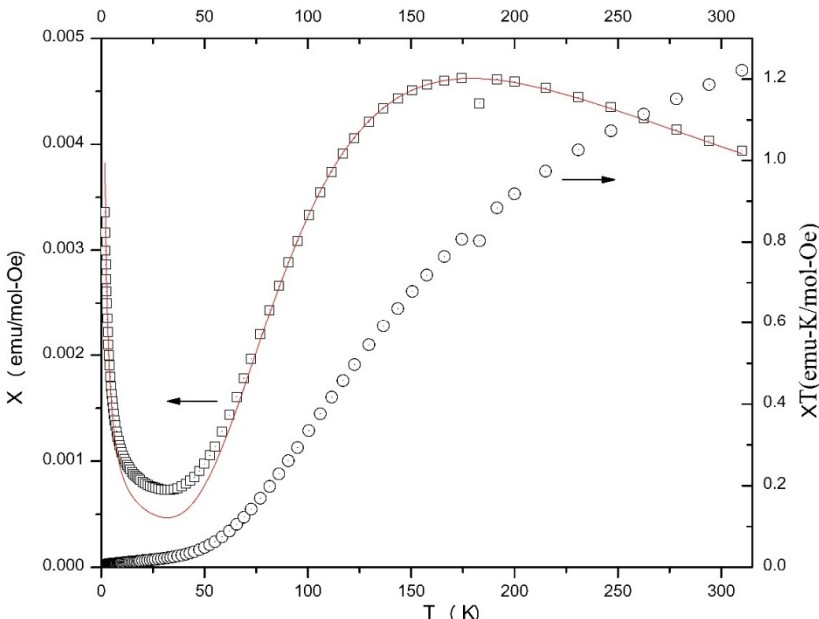

**Figure 9.** $\chi$ vs. *T* and $\chi T$ vs. *T* plots for compound **1**. Arrows on the plots indicate which *y*-axis. Applies to which data. The solid line is the best fit of the $\chi$ vs. *T* data to the *S* = 1/2 dimer model; see the text for details.

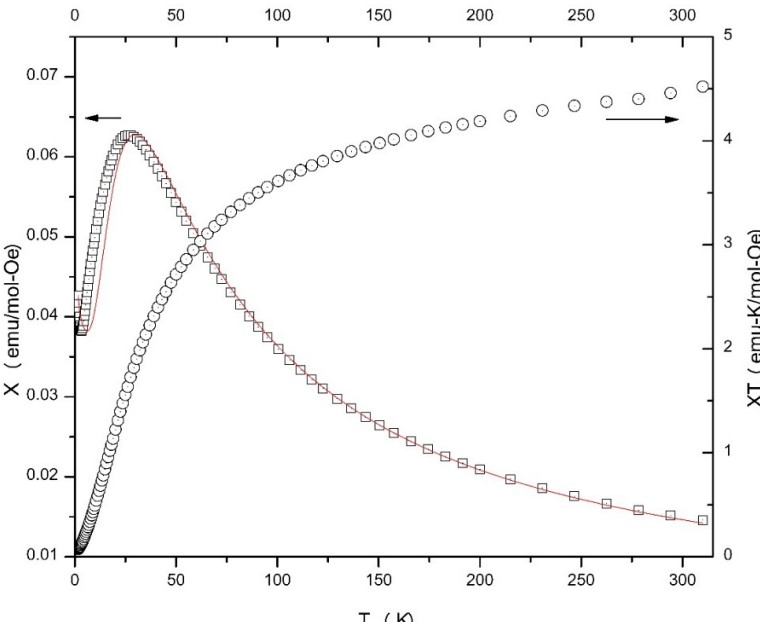

**Figure 10.** $\chi$ vs. *T* and $\chi T$ vs. *T* plots for compound **2**. Arrows on the plots indicate which *y*-axis applies to. Which data. The solid line is the best fit of the $\chi$ vs. *T* data to the cyclic *S* = 1 tetramer model; see the text for details.

Data for **1** show a maximum in $\chi$ near 175 K indicative of significant antiferromagnetic interaction (Figure 9). The $\chi T$ decreases as $T$ decreases reaching ~0 emu K/mol Oe at ~25 K. Magnetization for this complex reaches a saturation value of 36 emu/mol at 30 kOe, which is clearly too small for the bulk sample (Figure S22). A saturation value of nearly 24,000 emu/mol for four $Cu^{II}$ ions would be expected, indicating that the bulk sample is in a singlet ground state at 1.8 K and only a minor paramagnetic impurity is providing the observed moment. Susceptibility data for **1** were fit to a model for an $S = 1/2$ dimer using MAGMUN 4.1 [47]. The quality of fit is high through the maximum in $\chi$, but begins to deviate below 75 K, where the magnetism is dominated by the contributions of a trace magnetic impurity. The results yield a Curie constant ($C$) of 1.66(3) emu K/mol Oe ($0.41/Cu^{II}$), $J = -198(2)$ K (using the Hamiltonian $H = -J\Sigma S_1 \cdot S_2$) and $\rho = 0.4(1)\%$ ($\rho$ is the paramagnetic impurity). Attempts to fit the data to the same model with a Curie–Weiss correction to account for interdimer interactions yielded virtually identical results and a $\vartheta$ value of $-0.9(2)$ indicating that the interdimer interactions are not significant.

The symmetry and rectangular nature of the complex (Figure 1 and Figure S17) suggest a model with two $J$ values, one for the diazine (or hydrazonate)-mediated exchange and one for the alkoxido-mediated exchange. Attempts to fit the data to such a model with two $J$ values resulted in $J_1 = -195(2)$ K and $J = -1(1)$ K, again indicating that the exchange *between* the diazine-bridged dimeric units of the tetranuclear cation is negligible. This observation is in very good agreement with results for the structurally similar complex $[Cu_4(L)_4(H_2O)_2](NO_3)_4$ by the group of Thompson [14]. As in the case of the present complex **1**, the $Cu^{II}$ centers in the nitrate complex exist as diazine-bridged pairs of alkoxido-bridged dimers and they reported a value of ~ $-215$ K through the diazine bridge and 0 K through the oxygen bridge. The $J_2 = ~0$ K value is consistent with the orthogonal alkoxide bridging arrangement [14]. The $J_1$ value for **1** is entirely consistent with the large (155.3°) Cu-N-N-Cu torsion angle at the diazine bridge and the well established correlations involving this angle and exchange integral for a series of dinuclear copper(II) complexes [14,48,49] in which the metal ions are bridged by a diazine group. Thus, **1** can be considered as a practically isolated pair of antiferromagnetically coupled dinuclear units.

The magnetization of compound **2** reaches a value of ~1400 emu/mol at 50 kOe, well below the expected saturation value for four $Ni^{II}$ ions (~48,000 emu/mol), but the magnetization is still clearly rising (Figure S23) indicating that a significantly larger field would be needed to saturate the sample. This suggests the presence of measurable antiferromagnetic exchange in the complex, but also shows that the bulk material is still paramagnetic at 1.8 K, unlike sample **1**. The $\chi T$ product decreases as $T$ decreases rather smoothly in the 310–100 K range, and then more rapidly in the 100–1.8 K range reaching a value of ~0.05 emu K/mol Oe at 1.8 K.

Susceptibility data for **2** were fit to a model for an $S = 1$ cyclic (square) tetramer using MAGMUN 4.1 [47]. The fit is qualitatively acceptable, but overestimates the temperature of the maximum in $\chi$ by ~5 K, although the value of $\chi$ at $\chi_{max}$ is well reproduced (Figure 10). The fitted values are Curie constant = 4.61(3) emu K/mol Oe ($1.15/Ni^{II}$), $J = -12.6(1)$ K ($H = -J\Sigma S_1 \cdot S_2$), $\rho = 0.25(5)$ % and $D = 22.5(5)$ K. Attempts to fit the data to a simple $S = 1$ dimer model were unsuccessful. Although somewhat large, the fitted value for the single-ion anisotropy of the $Ni^{II}$ ions ($D = 22.5(5)$ K) is not without precedent [50–52] and substantially larger values have been reported [53–55]. It is clear that the structure has lost its solvate lattice molecules, based on the analytical data and on the powder X-ray diffraction pattern for the sample used for magnetic data collection, and as a result the symmetry of the system could have been reduced. The quality of fit suggests that the cyclic $S = 1$ model is reasonable, but the slight change in structure may render the superexchange pathways inequivalent and thus the values presented likely represent an average of $J$, $C$ and $D$ values. The inequivalence of the superexchange pathways might also be due to the different nature of the terminal donor atoms for the $Ni^{II}$ centers of the cation $[Ni_4(NO_3)_2(L)_4(H_2O)]^{2+}$. In any case, and especially in the absence of a better understanding of the structure of the desolvated material, additional detailed analysis would likely lead to an overinterpretation of the data available.

The antiferromagnetic *J* coupling between the Ni[II] centers in the cation of cluster **2** is clearly associated with the large Ni-O-Ni angles [7,39,56,57]; the Ni1-O1-Ni2, Ni2-O3-Ni4, Ni4-O4-Ni3 and Ni3-O2-Ni1 (Figure 3 and Figure S3) are 138.4(1), 138.2(1), 137.5(1) and 138.5(1)°, respectively.

## 3. Experimental Section

### 3.1. Materials, Physical and Spectroscopic Measurements

All manipulations were performed under aerobic conditions using reagents and solvents (Alfa Aesar, Karlsruhe, Germany; Aldrich, Tanfrichen, Germany) as received. The organic ligand LH was synthesized in typical yields of >95% as described in the literature [14–16,18], i.e., by the 1:1 reaction between picolinic acid hydrazide and 2-acetylpyridine in refluxing EtOH for 3 h. Its purity was checked by microanalyses (C, H, N), determination of the melting point (found, 193–194 °C; reported, 195–197 °C), and $^1$H NMR and IR spectra. Elemental analyses (C, H, N) were performed by the University of Patras (Patras, Greece) microanalytical service. Fourier transform infrared (FT–IR) spectra were recorded using a Perkin-Elmer 16PC spectrometer (Perkin-Elmer, Watham, MA, USA) with samples prepared as KBr pellets and as nujol or hexochlorobutadiene mulls between CsI disks. $^1$H NMR spectra of the diamagnetic Co(III) complexes were recorded on a 400 MHz Bruker Avance DPX spectrometer (Bruker, Karlsruhe, Germany) using (Me)$_4$Si as internal standard. UV/VIS solution spectra were recorded using a Specord 50 Plus spectrophotometer (Analytik Jena, Jena, Germany). Magnetic susceptibility data were collected using a Quantum Design MPMS-XL SQUID magnetometer (San Diego, CA, USA). Samples of **1** and **2** were ground and loaded into gelatin capsules. Magnetization data were collected as a function of field from 0 to 50 kOe at 1.8 K. Several data points were collected as the field was reduced back to zero to check for hysteresis effects; none were observed. Susceptibility data were collected for the background signal of the sample holder (measured independently), for the diamagnetic contributions of the constituent atoms as estimated via Pascal's constants [58], and for the temperature-independent paramagnetism of the Cu[II] and Ni[II] ions.

### 3.2. Synthesis of Complex [Cu$_4$Br$_2$(L)$_4$]Br$_2$·0.8H$_2$O·MeOH (**1**·0.8H$_2$O·MeOH)

To a stirred solution of LH (0.048 g, 0.20 mmol) in MeOH (20 mL) were added solids NaO$_2$CPh (0.029 g, 0.20 mmol) and CuBr$_2$ (0.045 g, 0.20 mmol). The resulting green slurry was stirred at room temperature for a further 30 min, filtered to remove an amount of NaBr and the green-brown filtrate was left undisturbed in a closed flask. X-ray quality, greenish brown crystals of the product were formed over a period of 3 days. The crystals were collected by filtration, washed with cold MeOH (2 × 1 mL) and Et$_2$O (3 × 2 mL), and dried in a vacuum dessicator over P$_4$O$_{10}$ overnight. The yield was 62%. The complex was satisfactorily analyzed as lattice solvent-free, i.e., as **1**. Analyses calculated for C$_{52}$H$_{44}$N$_{16}$O$_4$Cu$_4$Br$_4$ (found values in parentheses): C 40.79 (41.02), H 2.90 (2.84), N 14.64 (14.50) %. IR bands (KBr, cm$^{-1}$): 3080w, 3035w, 2960w, 1594m, 1576sh, 1534s, 1474s, 1436w, 1370s, 1322w, 1290m, 1258m, 1180m, 1144w, 1100w, 1080w, 1042m, 1016m, 920m, 804sh, 780m, 752m, 716m, 710m, 688m, 654sh, 640w, 574w, 565w, 504w, 462w. UV/VIS bands (MeOH, nm): 245, 280sh, 370, ~745.

### 3.3. Synthesis of Complex [Ni$_4$(NO$_3$)$_2$(L)$_4$(H$_2$O)](NO$_3$)$_2$·0.2H$_2$O·3EtOH (**2**·0.2H$_2$O·3EtOH)

To a stirred slurry of LH (0.048 g, 0.20 mmol) in CH$_2$Cl$_2$ (2 mL) was added a green solution of Ni(NO$_3$)$_2$·6H$_2$O (0.058 g, 0.20 mmol) in EtOH (10 mL). The resulting greenish brown solution was stirred at room temperature for a further 45 min, filtered and the filtrate was allowed to stand undisturbed in a closed flask. X-ray quality brown crystals of the product were precipitated over a period of two weeks. The crystals were collected by filtration, washed with cold EtOH (2 × 2 mL) and Et$_2$O (5 × 3 mL), and dried in a vacuum dessicator over anhydrous CaCl$_2$. Typical yields were in the range 30–35%. The complex was satisfactorily analyzed as lattice solvent-free, i.e., as **2**. Analyses calculated for C$_{52}$H$_{46}$N$_{20}$O$_{17}$Ni$_4$ (found values in parentheses): C 42.84 (42.67), H 3.19 (3.26), N 19.22 (18.87) %. IR bands (KBr, cm$^{-1}$): 3420mb, 3070w, 3030w, 2950w, 1598w, 1560w, 1522m, 1466m, 1438w,

1384s, 1368sh, 1294w, 1260w, 1182w, 1162w, 1144w, 1102w, 1066w, 1048w, 1026w, 922w, 810w, 780w, 761w, 714w, 694w, 642w, 562w, 505w, 420w. UV/VIS bands (MeOH, nm): 250sh, 290sh, 365, 615, ~980. The same complex can be prepared -in comparable yields- using 1 equiv. of $Et_3N$ per ligand in the reaction mixture.

### 3.4. Synthesis of Complex [Co₂(L)₃](ClO₄)₃·0.8H₂O·1.3MeOH (3·0.8H₂O·1.3MeOH)

To a stirred yellow solution of LH (0.024 g, 0.10 mmol) in MeOH (10 mL) was added solid $Co(ClO_4)_2 \cdot 6H_2O$ (0.037 g, 0.10 mmol). The solid soon dissolved and the resulting brown solution was stirred overnight at room temperature, filtered and the filtrate was allowed to stand undisturbed in a closed flask. X-ray quality brown crystals of the product were precipitated over a period of 3–4 days. The crystals were collected by filtration, washed with cold MeOH (1 mL) and $Et_2O$ (3 × 1 mL), and dried in air. The yield was 48% (based on the available LH). The complex was satisfactorily analyzed as [Co₂(L)₃]·H₂O, i.e., as **3**·H₂O. Analyses calculated for $C_{39}H_{35}N_{12}O_{16}CoCl_3$ (found values in parentheses): C 40.66 (40.54), H 3.07 (3.12), N 14.59 (14.74) %. IR bands (KBr, cm$^{-1}$): 3422mb, 3094w, 2951w, 2902w, 1606m, 1508m, 1458m, 1436m, 1376m, 1334w, 1300w, 1258w, 1178m, 1105sh, 1088s, 918w, 806w, 765sh, 756w, 718w, 688w, 662w, 624m, 510w, 412w. UV/VIS bands (MeCN, nm): 295, 395, 440, 580, 735. $^1H$ NMR peaks (DMSO-$d_6$, $\delta$/ppm): 8.59(d, 3H), 8.25(dd, 6H), 8.18(d, 3H), 8.02(d, 3H), 7.93(t, 3H), 7.55 (mt, 6H), 3.35(s, 9H), 3.17(s, 3H). The same complex can be prepared-in slightly higher yields(~55%)-by the addition of 1 equiv. of $Et_3N$ per ligand in the reaction mixture.

### 3.5. Syntheses of Complex [Co(L)₂](ClO₄)(4)

$Co(ClO_4)_2 \cdot 6H_2O$ (0.037 g, 0.10 mmol) and LH (0.072 g, 0.30 mmol) were dissolved in MeOH (9 mL). To the resulting red solution, $Et_3N$ (0.042 mL, 0.30 mmol) was slowly added. The solution became dark brown-red, was stirred overnight, filtered and left undisturbed in a closed flask. X-ray quality brown crystals of the product were precipitated over a period of 2–3 days. The crystals were collected by filtration, washed with cold MeOH (2 mL) and $Et_2O$ (10 × 2 mL), and dried in air. Typical yields were in the 50–55% range (based on the available cobalt). Analyses calculated. for $C_{26}H_{22}N_8O_6CoCl$ (found values in parentheses): C 49.03 (49.37), H 3.49 (3.45), N 17.60 (16.99) %. IR bands (KBr, cm$^{-1}$): 3074w, 3012w, 2956w, 2933w, 1602m, 1496s, 1474sh, 1450s, 1430sh, 1372s, 1328m, 1310w, 1292w, 1256w, 1170m, 1118s, 1092sh, 1080s, 994w, 916w, 812w, 782m, 772sh, 754w, 742w, 718m, 704m, 662w, 624m, 512w, 495w, 429w. UV/VIS bands (MeCN, nm): 285, 405, 450, 590, ~760. $^1H$ NMR peaks (DMSO-$d_6$, $\delta$/ppm): 8.58(d, 2H), 8.24(q, 4H), 8.15(d, 2H), 8.02(d, 2H), 7.88(t, 2H), 7.53 (mt, 4H), 3.34(s, 6H). Complex **4** can also be prepared by the 1:1 reaction between **3** and LH in refluxing MeOH in the absence of external base (yield <20%) and in the presence of LiOH (yield ~65%), Equations (6) and (7), respectively (*vide supra*).

### 3.6. Single-Crystal X-ray Crystallography

Suitable crystals of **1**·0.8H₂O·MeOH (0.09 × 0.13 × 0.18 mm), **2**·0.2H₂O·3EtOH (0.22 × 0.32 × 0.55 mm), **3**·0.8H₂O·1.3MeOH (0.07 × 0.12 × 0.17 mm) and **4** (0.19 × 0.29 × 0.30 mm) were taken from the mother liquor and immediately cooled to −113 °C (**3**·0.8H₂O·1.3MeOH) and −103 °C (for the other three compounds). Diffraction data were collected on a Rigaku R-AXIS SPIDER Image Plate diffractometer using graphite-monochromated Mo K$\alpha$ (**1**·0.8H₂O·MeOH, **2**·0.2H₂O·3EtOH) or Cu K$\alpha$ (**3**·0.8H₂O·1.3MeOH, **4**) radiation. Data collection ($\omega$-scans) and processing (cell refinement, data reduction and empirical absorption correction) were performed using the CrystalClear package [59]. The structures were solved by direct methods using SHELXS-97 [60] and refined by full-matrix least-squares techniques on $F^2$ with SHELXL, ver. 2014/6 [61]. Important crystallographic and refinement details are listed in Table S1. All non-H atoms were refined anisotropically. The H atoms of the four structures were either located by difference maps and refined isotropically or were introduced at calculated positions and refined as riding on their corresponding bonded atoms. Plots of the structures were drawn using the Diamond 3 program package [62].

The X-ray crystallographic data for the complexes have been deposited with CCDC (reference CCDC 1915158, 1915160, 1915159 and 1915157 for **1**·0.8H$_2$O·MeOH, **2**·0.2H$_2$O·3EtOH, **3**·0.8H$_2$O·1.3MeOH and **4**, respectively). They can be obtained free of charge at http://www.ccdc.cam. ac.uk/conts/retrieving.html or from the Cambridge Crystallographic Data Centre, 12 Union Road, Cambridge, CB2 1EZ, UK: Fax: +44-1223-336033; or e-mail: deposit@ccdc.cam.ac.uk.

## 4. Concluding Comments and Perspectives

It is rather difficult to conclude on a project that is still at its infancy. The present work extends the body of results (Table 5) that emphasize the ability of L$^-$ to form interesting structural types in 3d-metal chemistry. [2 × 2] rectangular (**1**) and square (**2**) grids have been characterized, while the interesting triple helicate-type dinuclear complex **3** was also isolated. Complexes **2–4** are the first, structurally characterized cobalt and nickel complexes of LH or L$^-$, while the two L$^-$ coordination modes in the Co(II) complexes (Table 5, Scheme 2) have been confirmed for the first time, emphasizing the flexibility and versatility of this ditopic ligand. The magnetic properties of **1** and **2** have been interpreted using one exchange interaction, and the former can be described as consisting of two antiferromagnetically coupled dinuclear units.

We believe that the research described herein has not exhausted new results. Indeed, studies in progress are producing additional products with other, magnetically interesting 3d-metal ions; our belief is that we have scratched only the surface of the coordination chemistry of LH/L$^-$. As far as future perspectives are concerned, we shall try to prepare lanthanide(III) clusters (only mononuclear complexes with the neutral ligand are known [17]; see also Table 5) and 3d/4f-metal complexes, based on L$^-$, with interesting magnetic properties. We are also trying to isolate complexes with ditopic ligands that are similar to LH, but with groups other than the methyl group (Scheme 1), because it is currently not evident whether the preparation and stability of 3d-metal complexes are dependent on the particular nature of the R substituent on the carbon atom next to the 2-pyridyl group.

**Supplementary Materials:** The following are available online at http://www.mdpi.com/: Figures S1–S4: IR spectra of the free ligand and representative complexes. Figures S5–S12: Solution UV/VIS/Near-IR electronic spectra of the complexes. Figures S13–S16: $^1$H NMR spectra of the diamagnetic Co(III) complexes in DMSO-d$_6$. Figure S17: ORTEP plot of the tetranuclear cation [Cu$_4$Br$_2$(L)$_4$]$^{2+}$. Figure S18: The 3D arrangement of complex **1**·0.8H$_2$O·MeOH. Figure S19: ORTEP plot of the tetranuclear cation [Ni$_4$(NO$_3$)$_2$(L)$_4$(H$_2$O)]$^{2+}$. Figure S20: ORTEP plot of the mononuclear cation [Co(L)$_2$]$^+$. Figure S21: Various structural plots of the cation [Co$_2$(L)$_3$]$^{3+}$. Figure S22: Magnetization data for complex **1** at 1.8 K. Figure S23: Magnetization data for complex **2** at 1.8 K. Table S1: Crystallographic data for the four complexes. Tables S2–S5: H-bonding interactions in the crystal structures of the complexes.

**Author Contributions:** E.P., E.S. and E.-K.M. contributed toward the syntheses, crystallization, and conventional characterization (IR, UV/VIS, $^1$H NMR) of the metal complexes. M.M.T. performed the magnetic measurements, interpreted the results and wrote the relevant part of the paper. C.P.R. and V.P. collected single-crystal X–ray crystallographic data, solved the structures and performed their refinement; the latter also studied in detail the supramolecular features of the crystal structures and wrote the relevant part of the article. S.P.P. coordinated the research, contributed to the interpretation of the results, and wrote the paper based on the reports of his collaborators. All the authors exchanged opinions concerning the interpretation and study of the results, and commented on the manuscript at all stages.

**Funding:** This research received no external funding.

**Acknowledgments:** Spyros P. Perlepes is grateful to the COST Action: CA15128-Molecular Spintronics (MOLSPIN) for encouraging his research activities in Patras.

**Conflicts of Interest:** The authors declare no conflict of interest.

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
