# Peer review of "Diversity of Coordination Modes in a Flexible Ditopic Ligand Containing 2-Pyridyl, Carbonyl and Hydrazone Functionalities: Mononuclear and Dinuclear Cobalt(III) Complexes, and Tetranuclear Copper(II) and Nickel(II) Clusters"

_magnetochemistry, doi:10.3390/magnetochemistry5030039_

Round 1

Reviewer 1 Report

This manuscript reports further chemistry of the N’-(1-(pyridin-2-yl)ethylidene)pyridine-2-carbohydrazide ligand, including the first Ni and Co complexes, all of which have been extensively characterized by X-ray crystallography, UV/vis and IR spectroscopy, elemental analysis, and magnetization measurements. It is a fitting homage to Prof. Yamashita and the introduction and chemistry sections were a particular pleasure to read. The reported complexes are interesting additions to the coordination chemistry repertoire and nicely demonstrate the coordination versatility of this flexible polytopic ligand. I recommend publication with the following changes/clarifications:

1) “The 1:1 reaction between Co(ClO4)2·6H2O and LH gives complex 3 according to Equation (4).”

Strictly speaking, equation 4 is a 2:3 reaction, not a 1:1 reaction. Then again, the compound is a 2:3 compound, so relative to the product, the reaction can be considered 1:1. However, if that convention is used, then equation 5 must also be considered a 1:1 reaction, based on the product stoichiometry. The authors should find a common convention, the simplest being referring only to reaction stoichiometry (2:3 for equation 4 and 1:2 for equation 5).

2) In the discussion of the structure of 2, “stripes” are mentioned, terminology which is unfamiliar to this reviewer. Do the authors mean “chains”, as in 2-D arrangements of molecules? The discussion in this part of the manuscript is quite dense and difficult to follow, and could be simplified.

3) It would be helpful to the reader to add the ChiT vs. T plots to the Chi vs T plots (on the same graph, using two y axes).

4) The magnetization given in emu/mol is quite unusual in this reviewer’s experience, and it would perhaps be better to report the data in the more common Bohr magneton per formula unit convention. Otherwise, this atypical presentation should be justified.

4) The analytical spectra (IR, UV/VIS) should be included as supplementary information.

Author Response

Reviewer 1

This manuscript reports further chemistry of the N’-(1-(pyridin-2-yl)ethylidene)pyridine-2-carbohydrazide ligand, including the first Ni and Co complexes, all of which have been extensively characterized by X-ray crystallography, UV/vis and IR spectroscopy, elemental analysis, and magnetization measurements. It is a fitting homage to Prof. Yamashita and the introduction and chemistry sections were a particular pleasure to read. The reported complexes are interesting additions to the coordination chemistry repertoire and nicely demonstrate the coordination versatility of this flexible polytopic ligand. I recommend publication with the following changes/clarifications:

We thank the reviewer for her/his comments.

1) “The 1:1 reaction between Co(ClO4)2·6H2O and LH gives complex 3 according to Equation (4).”

Strictly speaking, equation 4 is a 2:3 reaction, not a 1:1 reaction. Then again, the compound is a 2:3 compound, so relative to the product, the reaction can be considered 1:1. However, if that convention is used, then equation 5 must also be considered a 1:1 reaction, based on the product stoichiometry. The authors should find a common convention, the simplest being referring only to reaction stoichiometry (2:3 for equation 4 and 1:2 for equation 5).

The comment is absolutely correct. We use in the revised ms the simplest convention, referring only to reaction stoichiometry, as suggested by the reviewer.

2) In the discussion of the structure of 2, “stripes” are mentioned, terminology which is unfamiliar to this reviewer. Do the authors mean “chains”, as in 2-D arrangements of molecules? The discussion in this part of the manuscript is quite dense and difficult to follow, and could be simplified.

The comment is correct. We have replaced the term “stripes” by the “double chains” throughout the text of the revised ms, as well as in Table S3 of the revised “Supplementary Materials”. To simplify the discussion in the relevant part of the manuscript (as correctly requested by the referee), we have added a new explanatory sentence (“The donors of these H bonds……..oxygen atoms).

3) It would be helpful to the reader to add the ChiT vs. T plots to the Chi vs T plots (on the same graph, using two y axes).

We completely agree with the comment. The figures have been revised as requested, incorporating the χΤ vs. T data into the plots with two y-axes in the new Figures 9 and 10. Arrows in the plots indicate which y-axis applies to which data. Minor modifications in the text and in the captions of the new figures reflect this change.

  “4) The magnetization given in emu/mol is quite unusual in this reviewer’s experience, and it would perhaps be better to report the data in the more common Bohr magneton per formula unit convention. Otherwise, this atypical presentation should be justified.

While the reviewer is correct that magnetization data can be expressed as Bohr magnetons per formula unit, we disagree that presenting magnetization as emu/mol is “atypical”. In fact, we must also disagree that Bohr magnetons are more common. While that certainly was two or more decades ago, current literature uses units of emu/mol as commonly or more so. We are thus asking the reviewer’s and the Editor’s indulgence to retain the plots (Figures S22 and S23 in the revised “Supplementary Materials” section) as initially presented. However, to show that we fully respect the opinion of the reviewer, we have added a new sentence (“Although M vs. H per……. these units commonly”) in the first paragraph of part 2.4 of the revised ms, to indicate that both conventions can be used.

4) The analytical spectra (IR, UV/VIS) should be included as supplementary information.

Following the scientifically correct suggestion by the reviewer, we have incorporated the IR, UV/VIS/Near-IR and 1H NMR spectra in the revised “Supplementary Materials” section of our ms. The numbering scheme of the already existed figures has been modified accordingly. Minor changes in the text of the revised ms also reflect the incorporation of the new figures (Figures S1-S16).

WE ARE GRATEFUL TO THE REVIEWER FOR HER/HIS TIME TO STUDY OUR WORK AND FOR THE VALUABLE REVISION POINTS, COMMENTS AND SUGGESTIONS, WHICH HAVE HELPED US TO IMPROVE THE QUALITY AND READABILITY OF THE MS.

Reviewer 2 Report

Th eauthors report in an excellent written manuscript the diversity of coordination modes in a ditopc ligand consisting of 2-pyridyl, carbonyl and hydrazone functionality.

They synthesized and carefully characterized two tetranuclear clusters with Cu(II) and Co(II) centers along with a mononuclear and dinuclear Co(III) complexes.

The synthetic aspects, single crystal structures, spectroscopic (IR, UV/VIS, NMR) and magnetic investigations were performed and interpreted in a very professional manner.

I therefore recommend acceptance after very minor revisions/considerations:

The IR, UV/VIS and 1H NMR spectra should be included in the supplementary materials section too.

Author Response

REVIEWER 2

The authors report in an excellent written manuscript the diversity of coordination modes in a ditopic ligand consisting of 2-pyridyl, carbonyl and hydrazone functionality.

They synthesized and carefully characterized two tetranuclear clusters with Cu(II) and Co(II) centers along with a mononuclear and dinuclear Co(III) complexes.

The synthetic aspects, single crystal structures, spectroscopic (IR, UV/VIS, NMR) and magnetic investigations were performed and interpreted in a very professional manner.

I therefore recommend acceptance after very minor revisions/considerations:

We thank the reviewer for her/his warm comments.

The IR, UV/VIS and 1H NMR spectra should be included in the supplementary materials section too.

This is practically the same suggestion with the last comment of Reviewer 1. Following the scientifically correct suggestion by the reviewer, we have incorporated the IR, UV/VIS/Near-IR and 1H NMR spectra in the revised “Supplementary Materials” section of our ms. The numbering scheme of the already existed figures has been modified accordingly. Minor changes in the text of the revised ms also reflect the incorporation of the new figures (Figures S1-S16).

WE THANK THE REVIEWER VERY MUCH FOR HER/HIS TIME TO STUDY OUR WORK AND FOR THE VALUABLE REVISION SUGGESTION, WHICH HAS HELPED US TO IMPROVE THE QUALITY AND READABILITY OF THE MS.